# An agent-based model to advance the science of collaborative learning health systems

Michael Seid[1,2,3]*, David Bridgeland[4], Christine L. Schuler[1,5], David M. Hartley[1,3]

**1** Department of Pediatrics, University of Cincinnati College of Medicine, Cincinnati, Ohio, United States of America, **2** Division of Pulmonary Medicine, Cincinnati Children's Hospital Medical Center, Cincinnati, Ohio, United States of America, **3** James M. Anderson Center for Health Systems Excellence, Cincinnati Children's Hospital Medical Center, Cincinnati, Ohio, United States of America, **4** Hanging Steel Productions, LLC, Island, Virginia, United States of America, **5** Division of Hospital Medicine, Cincinnati Children's Hospital Medical Center, Cincinnati, Ohio, United States of America

* michael.seid@cchmc.org

## Abstract

Improving the healthcare system is a persistent and pressing challenge. Collaborative Learning Health Systems, or Learning Health Networks (LHNs), are a novel, replicable organizational form in healthcare delivery that show substantial promise for improving health outcomes. To realize that promise requires a scientific understanding that can serve LHNs' improvement and scaling. We translated social and organizational theories of collaboration to a computational (agent-based) model to develop a computer simulation of an LHN and demonstrate the potential of this new tool for advancing the science of LHNs. Model sensitivity analysis showed a small number of parameters with outsized effect on outcomes. Contour plots of these influential parameters allow exploration of alternative strategies for maximizing model outcomes of interest. A simulated trial of two common health system interventions – pre-visit planning and use of a registry – suggested that the efficacy of these could depend on LHN current state. By translating heuristic theories of LHNs to a specifiable, reproducible, and explicit model, this research advances the scientific study of LHNs using tools available from complex systems science.

## Significance statement

Improving the US healthcare system is a persistent and pressing challenge. Learning Health Networks (LHNs) are a novel organizational form in healthcare delivery that show substantial promise for improving health outcomes. Until now, approaches for understanding and advancing LHNs have been descriptive and empirical. We translated social and organizational theories of collaboration to an agent-based model to develop a computer simulation of an LHN and demonstrate the potential of modeling and simulation for advancing the science of LHNs. Such an approach could advance

**Data availability statement:** All relevant data are within the manuscript and its Supporting Information files. This includes the implementation of the model in Python and the complete list of parameter values and initial conditions that are necessary for reproducing the results depicted in the study. The model is fully stochastic and therefore slight statistical variation may be observed.

**Funding:** Funding for this project came from an internal grant from Cincinnati Children's Hospital Medical Center (no number) and from the State of Ohio, Ohio Development Services Agency, Ohio Third Frontier, Grant Control No. TECG2022-1691. The content of this publication reflects the views of the authors and does not purport to reflect the views of the Ohio Development Services Agency. The funders had no role in study design, data collection and analysis, decision to publish, or preparation of the manuscript.

**Competing interests:** I have read the journal's policy and the authors of this manuscript have the following competing interests: MS and DMH are co-inventors on "A computation model of learning networks." Assignee: Cincinnati Children's Hospital. US Patent application filed 5/5/21 number: 17/291,401]. This does not alter our adherence to PLOS ONE policies on sharing data and materials.

the growth and optimization of LHNs and, thus, more rapidly and reliably improve healthcare delivery and outcomes.

## Introduction

Improving the US healthcare system is a persistent and pressing challenge. Learning Health Networks (LHNs) [1], a type of collaborative learning health system [1,2], have shown improved health outcomes in chronic conditions [3–5], safety [6], and community health [7], but their small scale limits their impact. Increasing LHN scale and impact requires a better understanding of the phenomenon (e.g., How do LHNs work? What are the mechanisms of action?), as well as actionable insights (e.g., Given the current state of a LHN, what intervention(s) might improve its functioning?). While mid-range [8] heuristic theory [1,9–12] and experience [13–17] have accumulated, these are descriptive, rather than explanatory. Without a specifiable, reproducible, and explicit model of LHNs to advance understanding and generate actionable insights, improving and scaling LHNs will be slow, scattershot, and unsuccessful.

In this study, we take the first steps in addressing this gap by translating mid-range LHN theory into a computational model so as to ready the field for further studies and model validation. We summarize the mid-range theory of LHNs as complex systems that use an actor-oriented organizational architecture to facilitate networked coproduction. We translate that theory into an agent-based model and describe the model and underlying logic. We examine the model by computationally implementing a set of theoretically relevant parameters and by illustrating an approach to simulating the effect of different interventions on possible future states for a LHN, given LHN current state.

### LHNs as complex systems that re-organize healthcare for collaboration

It has been suggested that healthcare in general [18], and LHNs in particular [1], are complex adaptive systems – collections of individual agents that interact in such a way as to change the context for other agents. Complex adaptive systems are characterized by resilience, emergence, and self-organization. In complex adaptive systems, agents act with respect to achieving goals or objectives. In healthcare, agents' (organizations', individuals') goals or objectives include achieving best possible health outcomes.

Improving health outcomes requires getting the right treatment to the right person at the right time, every time. While seemingly simple, the sheer number of conditions and treatments (leaving aside comorbidities, social determinants of health, and combinations of treatments) makes this matching extremely difficult. Moreover, a patient's needs and goals might change over time, as might their illness. At an individual level, doctors and patients must identify appropriate treatment options, evaluate the degree to which chosen treatments improve outcomes that the patient cares about, and use the information from that evaluation to change treatments if necessary. To do this, doctors and patients act and interact repeatedly, over time, coproducing [11,19] the

information (e.g., symptom reports), knowledge (e.g., differential diagnosis, treatment options), and know-how (e.g., dealing with side-effects) required to make and improve the match.

In the above individual-level case, a patient and a doctor could match treatments to conditions using only their own information, knowledge, and know-how. However, solving this matching problem in silos, without accumulating, using, and sharing knowledge about what might work and making changes as necessary, is inefficient and ineffective. While healthcare organizations use a variety of means to accumulate and share knowledge (e.g., clinical trials, an electronic health record), ensuring the right match at an organizational level requires rapid and flexible mobilization and deployment of appropriate information, knowledge, and know-how; ways to facilitate problem-solving and collaboration; and flexibility so that agents can act to improve outcomes. In an organization that is fragmented, inflexible, and/or hierarchical, this is difficult or impossible. What is required, instead, is for the organization to have a) sufficient actors (e.g., patients, clinicians, organizations) willing and able to self-organize towards a common goal; b) a commons where the actors create and share resources (information, knowledge, and know-how); and c) infrastructures, processes, and protocols that facilitate multi-actor collaboration. This organizational form, novel in health care delivery [1,15,20], is known as an actor-oriented architecture (AOA) [12,21]. LHNs use the AOA scheme to facilitate coproduction and collaboration around improving health outcomes.

## Agent-based models as a tool for understanding and insight

Agent-based models (ABMs), in which populations of agents with various attributes interact according to rules and in environments specified by the modeler, have been used to gain insights into a variety of complex systems and emergent phenomena [22] including multi-cellular systems in biology [23], the COVID-19 Pandemic [24], flood risk management [25], autonomous vehicles [26], new product market diffusion [27], and economic and social behavior [28]. ABMs have also been used to understand healthcare as a complex adaptive system [29–31]. Complex adaptive systems in general, and actor-oriented organizations like LHNs specifically, are theorized to be dependent on how actors interact. Because agent-based models simulate the system level effect of many local interactions, they are well-suited as a modeling tool for complex adaptive systems such as LHNs. If a computational model can produce LHN-like behavior based on agent interactions, it likewise suggests that LHNs can be thought of as complex adaptive systems not only heuristically, but functionally as well.

An important use of ABMs is to help transform heuristic theories into scientific ones [32]. Heuristic theories are typically communicated descriptively, as in the previous section. Many theories, especially in social and organizational sciences, are expressed heuristically, which creates limitations [33]. Plausible heuristic theories may contain unexamined assumptions or concepts that are vague and subject to different interpretations. They may be overly general or ignore context. This impedes conceptual clarity and scientific understanding, slowing the growth of knowledge and its application in the real world. Mathematical models such as ABMs force scientists to translate vague concepts into formal logic that is coded and programmed [33]. Making explicit the assumptions underlying the phenomenon of LHNs serves several purposes. First, it imposes discipline on the discourse: Constructs and the relationships among them must be well-specified. Second, it allows exploration of how the phenomenon of LHNs might emerge out of local conditions. Third, it can "illuminate core dynamics" [34] of LHNs and lend insight into mechanisms of action. Having a formal logic and making assumptions explicit is even more important when, as in the case of learning health systems, the field is inherently interdisciplinary [33,35].

ABMs, and models generally, are useful for a variety of purpose [34]. While models can be used for point prediction, here our goal is at once broader and more modest. It is to translate existing theories of LHNs to a specifiable, reproducible, and explicit model. Empirical calibration is an important validation-supporting process that will be addressed in future work.

**What this study adds**

While the above mid-range, heuristic theories of coproduction of care and the AOA describe agent motivations and behaviors and the organizational form of a LHN, critical gaps in our knowledge remain. First, while theory [19] suggests that good care is the result of productive interactions between prepared, proactive clinicians and active, engaged patients, how does coproduction of care arise from agent actions and interactions? Second, descriptive accounts suggest that LHNs use a set of social [9,36] and technological [37] infrastructures and implement a set of processes [38] over several phases of development to achieve an actor-orientation [15,39]. How do LHNs become, or become more, actor-oriented? This study begins to close these gaps in our knowledge by translating these heuristic theories into a specifiable, reproducible, and explicit model.

## Materials and methods

### Model

The full model description, following the ODD (Overview, Design concepts, Details) protocol for describing individual- and agent-based model [40], as updated by Grimm et al [41], is available as S1 Supplement. The following is excerpted and simplified from that Supplement.

**Purpose and patterns.**  The purpose of the LHN ABM is to establish and assess hypotheses about the general behavior of LHNs under a range of initial starting conditions and configurations, and in consideration of key hypothesized mechanisms of action. The model is designed to enable careful thinking about how and why LHNs achieve results. The current version of the model is not meant to predict future states: It is meant to help LHN leaders to think carefully about strategies and tactics, and researchers to make better-designed experiments.

The LHN ABM models the change in health status over time for a population of patients as a result of patient and clinician interactions under conditions thought to be important to the functioning of a LHN. The model is meant to produce several patterns suggested by the theory of LHNs. In general, as patient and clinician engagement, influence among patients and clinicians, and ease of knowledge creation and sharing increases, so should the amount of knowledge available for treatment decision making and the health of the patient population increase.

**Agents.**  *Patients* are agents representing an individual with the given condition. Patients are assigned to a care center, and within a care center, to a clinician. Patient health status can vary from 0 (worst) to 1 (best). Each patient has a phenotype, denoting the degree to which the patient will respond to each of the available treatments. Each patient has phenotype response information, or how much is known about the treatment of the medical condition assigned to the patient, and individual response information, or how much the patient knows about their own response to treatments. Patients have a state variable of level of engagement with the LHN. Engagement is an ordinal scale including: unaware, aware, participating, contributing and owning. *Clinicians* are agents representing treating individuals. Clinician states include engagement, with the same ordinal scale as for patients. Clinicians are assigned to a care center and have a panel of patients.

**Basic principles.**  The theory on which the LHN ABM is based is that LHNs use an AOA to facilitate collaboration and coproduction [1,21,39]. That is, LHNs must bring together sufficient actors (in the ABM, patients and clinicians) with the will and ability (in the ABM, their level of engagement) to self-organize. They must have a commons where actors create and share resources (in the ABM, the commons and the Enhanced Registry [37]), and processes, protocols, and infrastructure to facilitate multi-actor collaboration (in the ABM, the clinical encounter, patient and clinician influence, and the degree to which patients and clinicians create, share, and use common resources). With these in place, we hypothesize that there ought to be better coproduction [11] (in the ABM, praxis, explained below) between prepared proactive clinicians and active, involved patients [19], and thus, the match between needs and treatments should be optimized, leading to better health outcomes.

**Adaptation.** Patients may leave the cohort if they age out, move away, get sufficiently better, or get sufficiently sicker. Patients and clinicians may become more engaged if they interact with another agent (clinician or patient) with a higher level of engagement. Patients may become less engaged if they interact with a clinician with a lower level of engagement. Clinicians may become less engaged periodically (burnout). Patients and clinicians can contribute some amount of knowledge at some periodicity to the commons, depending on their level of engagement.

Direct objective-seeking behaviors occur in the context of the clinical encounter. At the first clinical encounter, a treatment is chosen from the full set of available treatments. At subsequent clinical encounters, the patient may change treatments if the patient's perceived health status has not improved since the last clinical encounter. If the patient changes treatments, the alternatives are chosen from the set of available treatments excluding the current treatment. The input driving treatment selection is selection efficiency, which influences the likelihood that the best (and second best, and third, best, etc.) treatment for that patient's phenotype is chosen from among the alternatives.

**Objectives.** The objective is improved health status via matching the patient's phenotype to the best treatment for that phenotype. A correct match is one of several influences (along with relapse, condition natural history, condition variability, and treatment effectiveness) that influences patient health status. The rationale for calculating health status based on selection of the most effective treatment for that phenotype and disease characteristics is based on general, albeit simplified, principles of clinical medicine.

**Learning.** The adaptive behavior of patients and clinicians—deciding on a treatment—is modeled using an approach that includes learning, based on the theory of reasoned action [42]. In this case, patients and clinicians learn from the current treatment's effect on patient health status (individual response information), as a patient may bring information to the clinical encounter about the effect of the current treatment. They also learn from information about patients with the same phenotype (phenotype response information) based on resources shared in the commons.

**Prediction.** The adaptive behavior of treatment selection is based on an explicit prediction that each treatment has a certain probability of improving the patient's health status. Prediction is modeled to represent how patients and clinicians actually make predictions. That is, if the patient's health status is not perceived as improved, a treatment will be selected from the other available treatments, based on the degree to which patients and clinicians are able to co-create and use knowledge for treatment selection. In the model, this use is referred to as selection efficiency.

**Interaction.** There are three kinds of direct interactions between agents in the model: Interactions within a clinical encounter between patients and clinicians, interactions among patients, and interactions among clinicians. There are also indirect interactions, as when patients or clinicians contribute knowledge to the commons.

Within clinical encounters, as above, patients and clinicians share knowledge for the purpose of decision making. Patients and clinicians can also affect each other's level of engagement. A patient who is unaware at a particular time period can become aware as a result of a clinical encounter. If an unaware patient has a clinical encounter with a clinician who is not unaware (i.e., aware, participating, contributing, or owning), the patient might become aware of the learning network. A clinical encounter between a patient who is less engaged and a clinician who is more engaged can cause an increase in patient engagement. A clinical encounter between a patient who is more engaged and a clinician who is less engaged can cause a decline in patient engagement.

Interactions among patients and among clinicians can also change engagement levels. Each patient has influence connections to some other patients. Over time, an influence link between an unaware patient and a patient who is at least aware can result in the unaware patient becoming aware of the learning network. An influence link can also influence a patient who is already aware of the learning network to a greater level of engagement (i.e., from aware to participating, from participating to contributing, or from contributing to owning). This is modeled in the same way for clinicians.

At each time step, patients and clinicians who are sufficiently engaged might contribute items to the commons. Other patients and their clinicians, if sufficiently engaged, might use those items (or other items in the commons) to increase phenotype response information or individual response information.

**Emergence.** The model's primary results—number of items in the commons, praxis (knowledge available for treatment decision making), and patient health status—emerge from patient and clinician engagement, patient and clinician influence, care center states, characteristics of the condition being treated, and the commons. Praxis, generally, refers to a process by which a theory, lesson, or skill is enacted or applied. In this case, we use it to mean knowledge applied to treatment decision making.

**Stochasticity.** The model uses stochasticity to model initial conditions, to simplify the modeling of subprocesses, and to reflect the observed choices of actual agents (see S1, Stochasticity, p 7–9). Incorporating stochasticity at each time-step of the model reflects the role of chance in living systems and makes it necessary to analyze ensembles of model runs.

**Influence diagram.** The influence diagram for the LHN ABM is shown in Fig 1. Starting on the right-hand side of the Figure, a patient's health status is perceived by the clinician and patient during a clinical encounter (Arrow 1), with some degree of evaluation accuracy (Arrow 2). This, and other information from the clinical encounter is knowledge about treatment response that might be shared in the commons (Arrow 3). Based on perceived health status, there may be a change in treatment during a clinical encounter (Arrow 4). The selection of an effective treatment regimen (Arrow 5) depends on the selection efficiency of the clinician and patient – their ability to prefer an effective treatment regimen instead of an ineffective one (Arrow 6). Both selection efficiency and evaluation accuracy are driven by praxis (Arrows 7 and 8). Praxis is the result of both phenotype response information—information about the phenotype of the patient, typically from the commons (Arrow 9)—and individual response information—information about how well the patient is responding to the treatment regimen, which is affected by the level of engagement [30] of the patient (Arrow 10) and of the clinician (Arrow 11) and the amount of shared knowledge in the commons. Shared knowledge accumulates in the commons as patients (Arrow 12) and clinicians (Arrow 13) contribute knowledge, depending on their level of engagement. Patients (Arrow 14) and clinicians (Arrow 15) can, depending on their level of engagement, access phenotype or individual response information from the commons. The engagement of each patient can change over the course of the simulation, as a result of experiences during the clinical encounter (Arrow 16), and by connections to other patients (Arrow 17). The engagement of each clinician also changes over the course of the simulation, affected by connections with other clinicians (Arrow 18).

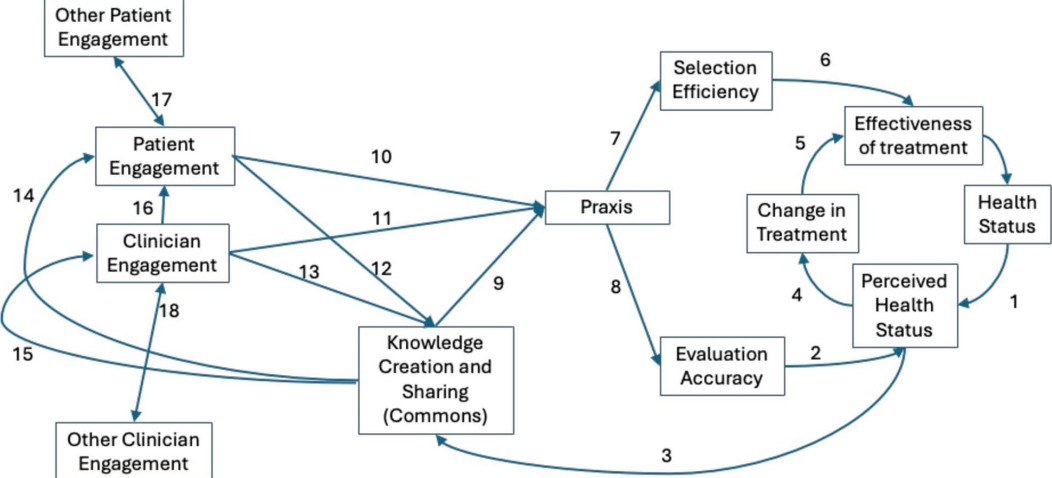

**Fig 1. High-level influence diagram for the LHN ABM.**

**LHN-level interventions.** Pre-visit planning (PVP) and an enhanced registry (ER) are characteristics of the care center. These were modeled as LHN-level interventions because they are widely applicable across conditions and a focus of many existing LHNs.

PVP is a cornerstone of Wagner's Chronic Care Model [19], enabling prepared, proactive clinicians and active, engaged patient to have productive interactions. PVP can range from simple shared agenda-setting (asking patients what they'd like to talk about in the clinical encounter) to sophisticated clinical decision-support for precision medicine based on genotype, phenotype, past and current treatments and outcomes. The effect of PVP in our model is represented by its impact on engagement level (e.g., having the right information at hand makes it easier for clinicians to be more collaborative), on the amount of information brought to the clinical encounter by patients and clinicians, and by praxis (e.g., sophisticated clinical decision-making tools from PVP will prompt and guide treatment decisions in the clinical encounter). We created a parameter to represent three levels of PVP sophistication – low, medium, and high. PVP functions mathematically by constraining effectiveness of individual response information and phenotype response information. In the model, the PVP function saturates. If PVP is set to high, the maximum effectiveness of individual response information and phenotype response information is set to a unit level. If PVP is set to medium or low, the maximum effectiveness is lower.

An ER represents a care center's ability to upload, aggregate, and use clinical data collected as part of care. This capability in which data captured in the course of clinical care is available to the registry in more or less real time and by an architecture that supports clinical care, quality improvement, and research is a so-called triple-use registry [37]. Implementation of an ER allows data to be available for pre-visit planning, facilitates data aggregation and analysis, and allows knowledge and insights to be shared more seamlessly. The effect of an ER in our model is represented by its impact on PVP and on knowledge sharing. Enhanced Registry has an ordinal scale: 'low' denotes a care center in which data are not uploaded automatically, 'medium' denotes a care center that uploads clinical data automatically to the registry but without the ability to do real time PVP, and 'high' denotes a care center that uploads clinical data and has it available for use at the clinical encounter.

**Translation of heuristic theory to ABM parameters.** The theoretical elements comprising the Chronic Care Model and the AOA form the basis of a set of processes that LHNs typically put into place [15]. The processes are organized, in a Network Maturity Grid [38], across several domains including systems of leadership, governance and management, quality improvement, engagement and community building, data and analytics, and research. The theoretical elements also underly several of the main change concepts typically implemented in LHNs [15,39]. Table 1 shows how the theoretical elements are instantiated in LHNs through process maturation across Network Maturity Grid domains [38], examples of changes typically implemented to create LHNs, and how these are represented, via specific parameters, in the ABM (definitions of these parameters are available in S2 Supplement).

We attempted to design parameters that could parsimoniously describe variation in LHNs relevant to the heuristic theory of the chronic care model and the actor-oriented organizational architecture. These 30 parameters are only a subset of all the possible influences on an LHN or of all the ways that an LHN could vary. We did not include variables characterizing broader macro-systems, such as cost-effectiveness population demographics, or payor mix.

## Approach

Approach to validation: Following Collins et al, here we define validation as the process of determining if a model adequately represents the system under study for the model's intended purpose [43]. The model's intended purpose at this stage is to establish and assess hypotheses, based on existing heuristic theories, about the general behavior of LHNs in consideration of key theorized mechanisms of action, and under a range of initial starting conditions and configurations. Therefore, we approached validation by a) determining the degree to which a parsimonious set of theoretically important parameters are related to model outcomes of interest and b) how those parameters could be used to develop hypotheses about LHN behavior. Further, we c) illustrate how the model could inform an approach for improving LHN outcomes. We

**Table 1. LHN theoretical elements, Network Maturity Grid domains, LHN change concepts and names of parameters in LHN ABM.**

| Theoretical Elements | NMG Domain | LHN Change Concept | Names of parameters in LHN ABM (See S2 Supplement for definitions) |
|---|---|---|---|
| Chronic Care Model, Coproduction | Data and analytics, Quality Improvement, Engagement and community building | Implement all six aspects of the Chronic Care Model | 1) encounter period, 2) encounter aware determiner, 3) patient activate determiner, 4) patient dispirit determiner, 27) selection efficiency maximum, 28) evaluation accuracy minimum praxis, 29) patient engagement degree participating, 30) clinician engagement degree participating. |
| AOA – Sufficient numbers of actors with the values and skills to self-organize | Systems of leadership Engagement and community building Governance and management | Leadership to align all participants around a shared goal and to build a culture of generosity and collaboration | 5) patient network edges, 6) patient influence across prop,7) patient influence become aware probability, 8) patient influence activation probability, 9) clinician network edges 10) clinician influence across prop, 11) clinician influence become aware probability, 12) clinician influence activation probability, 13) clinician dispirit probability |
| AOA – A commons where actors create and share resources | Data and analytics Engagement and community building | Platforms for creating and sharing common resources | 14) shared knowledge initial, 15) shared knowledge half-life, 18) enhanced registry initial per patient, 19) enhanced registry analysis period, 20) enhanced registry record per commons item, 23) patient response info half-life |
| AOA – Processes, protocols and structures that make it easier to form functional teams | Governance and management Quality Improvement Engagement and community building Data and analytics | Network governance policies that facilitate sharing Data registries that support clinical care, improvement, and research | 16) patient shared knowledge contrib determiner, 17) clinician shared knowlede contrib determiner, 21) potential phenotype response info from SK unit, 22) phenotype realization numeric, 24) patient response info increase numeric, 25) patient response info acceleration from SK unit, 26) maximal patient response info acceleration from SK |

Theoretical Elements of LHNs [1], Network Maturity Grid (NMG) Domains [38], LHN Change Concepts, and associated Parameters in the LHN ABM. Theoretical elements include the Chronic Care Model [19] and coproduction [11], and the Actor-Oriented Architecture (AOA) [21]. LHS Change Concepts have been shown to be common across existing LHNs [15,39]. The parameters associated with these in the LHN ABM can be manipulated by the modeler, and outcomes across different initial settings can be compared.

pursued these goals via a) sensitivity analysis to determine important parameters, b) a solution space exploration based on the subset of the most important parameters, and c) simulated experiments on the effect of PVP and ER in LHNs that vary on evaluation accuracy and selection efficiency. All simulations represent five years of the scenario, which we found to be sufficient to achieve a steady state, and a closed cohort (i.e., no new patients, no patients leave), to simplify cohort effects. All syntax is included in S3 Supplement.

The model was written in Python version 3.12.4 utilizing the following libraries: pandas (version 2.2.2), scipy (1.14.0), transitions (0.9.1), mesa (2.3.2) [44]. The PRCC analysis was performed using R, version 4.5.1 utilizing the sensitivity library.

**Sensitivity analysis.** Sensitivity analysis is useful for understanding which elements, or combinations of elements, of a model have the greatest impact on results, as well as how various elements interact [45]. This is essential for model interpretability: If all or most model elements have similar impact on the outcomes of interest, little insight is gained for simplifying the model.

Parameters included in the sensitivity analysis were developed from the theoretical elements of LHNs, as described above and in Table 1. Other parameters in the model were developed to enable simulation of different health conditions (e.g., number of phenotypes, number of treatments, severity, course, relapse probability) or to simulate different initial conditions (e.g., patients joining the network over time vs already all enrolled, patients entering and leaving the cohort vs closed cohort). These parameters were held constant in the sensitivity analysis. The 30 parameters included in the sensitivity analysis are further specified in S2 Supplement.

We considered three model outputs for the sensitivity analysis: cumulative average change in health status (health), cumulative average change in praxis (praxis), and cumulative increase in number of items in the commons (knowledge).

We examined health as an outcome because improving health is the purpose of LHNs. Praxis was chosen because it is a key variable in the learning, prediction, and sensing components of the model. In the ABM, praxis influences the accuracy with which the patient and clinician evaluate the patient's current health (and therefore to correctly decide whether to change treatments) and the efficiency by which an alternate treatment is selected. Knowledge was chosen because praxis is based, in part, on the degree to which phenotype response information and individual response information are brought to the clinical encounter. A key theory about LHNs is that patients and clinicians contribute to and use knowledge available in the commons to increase phenotype response information and individual response information.

We executed a sensitivity analysis using the Latin hypercube sampling-partial rank correlation coefficient (LHS-PRCC) approach to identify the most influential model parameters for each of the three outcomes considered [46–48]. Using Latin hypercube sampling, random samples from uniform or triangle distributions of the 30 model parameters were generated. Model outputs corresponding to this set of parameters were obtained and PRCCs were calculated using the sensitivity package [49] in R version 4.3.3 [50]. PRCCs take on values between −1 and +1; PRCCs with absolute values close to unity indicating strong impact on the model output. Trials demonstrated PRCC stability for 800–900 replicates; we used 1000 replicates for the results described below.

**Solution space exploration.** To test the degree to which the ABM LHN could be used to develop hypotheses about LHN behavior, we created contour plots to show the functional relationship between each of the three outcomes of interest (health, praxis, knowledge), and the two most important model parameters for each (as determined via the LHS-PRCC). For each outcome, we divided each of the two parameters into 41 equal increments to create an XY grid and plotted the outcome of these 1,681 runs on the Z axis (height). All other parameters were held constant at their midpoints. Contour lines joined points of equal Z value to represent Z values of constant height. The resulting contour plots represent the three-dimensional space that allows visualization, for any combination of the two parameters of not only the Z value at that point, but also the 'terrain' between the current Z value and the highest Z value.

**Simulated experiment.** With an understanding of which parameters are influential and how they relate to outcomes of interest, we illustrate how the model could inform LHN strategy and optimization. In this illustration, we imagine LHNs considering whether and to what degree to implement PVP and/or ER. In this case, their choices are to implement a low, medium, or high level of PVP and/or ER. We further imagine that not all LHNs are the same. Specifically, we simulate the implementation of PVP and ER in three different LHNs that are characterized by two parameters influential in health status improvement: evaluation accuracy and selection efficiency.

We used the contour plot in Fig 4a to select the three LHNs. The 'low' functioning LHN is defined by a point at the bottom right part of Fig 4a. To do this, we set selection efficiency at 10 and evaluation accuracy (reversed) at 0.20. As seen in Fig 4a, this low functioning LHN is producing, on average, −0.1 change in health status. The 'medium' LHN comes from the central part of Fig 4a. In this case we set selection efficiency to 200 and evaluation accuracy (reversed) at 0.10. Per Fig 4a, this LHN is producing, on average 0 change in health status. The 'high' functioning LHN comes from the upper left part of Fig 4a, with selection efficiency of 1000 and evaluation accuracy (reversed) at 0.001. As seen in Fig 4a, this LHN is producing, on average, 0.15 change in health status.

The simulated experiment is, therefore, a 3X3X3 factorial experiment of PVP (low, medium, high) and ER (low, medium, high) in the context of three different LHNs (low, medium, and high), with the outcome being median patient health status (Table 2).

**Table 2. Parameter specifications for simulated experiment.**

| Parameter | Low | Medium | High |
|---|---|---|---|
| LHN Functioning (settings for selection efficiency and evaluation accuracy that result in LHN effect on health status of…) | sel eff: 10<br>eval acc: 0.001 | sel eff: 100<br>eval acc: 0.01 | sel eff: 1000<br>eval acc: 0.01 |
| Pre-visit Planning (the maximum possible effective individual response information and the phenotype response information) | 0.2 | 0.5 | 1.0 |
| Enhanced Registry (an ordinal scale denoting on/off for automated data upload and real-time PVP) | No automated data upload | Automated upload but no real-time PVP | Automated upload and real-time PVP |

## Results

### Sensitivity analysis

Fig 2 shows the PRCCs for health (Fig 2a), praxis (Fig 2b), and knowledge (Fig 2c). The two largest (absolute value) PRCCs for health are selection_efficiency_maximum (PRCC = 0.81), which is the maximum selection efficiency for a clinician with maximum praxis of 1.0, and evaluation_accuracy_minimum_praxis (PRCC = −0.69), which is the minimum evaluation accuracy in a clinical encounter for a clinician with zero praxis (reversed). The two largest PRCCs for praxis are clinician_engagement_degree_participating (PRCC = 0.61) and patient_engagement_degree_participating (PRCC = 0.58). These refer to the amount of phenotype response information and individual response information accessed by clinicians and patients, respectively, at various levels of engagement. The two largest PRCCs for the commons are shared_ knowledge_half-life (PRCC = 0.70), which is the rate of decay of knowledge in the commons (e.g., how quickly knowledge goes out of date), and patient_influence_activation_ probability (PRCC = 0.67), which is the probability (annual) that a more highly activated patient will cause a less activated patient to become more activated. Fig 3 shows the high-level influence diagram with the arrows corresponding to the larges PRCCs thickened (shared_knowledge_half-life not shown).

### Solution space exploration

Fig 4 shows the contour plot for health status as a function of selection_efficiency_maximum and evaluation_ accuracy_minimum_praxis (Fig 4a), for praxis as a function of clinician_engagement_degree_participating and patient_ engagement_degree_participating (Fig 4b), and knowledge as a function of shared_knowledge_half-life and patient_influ- ence_activation_ probability (Fig 4c). As can be seen, the 'topography' of these plots differs across outcomes, with the area between changes in elevation analogous to the steepness of the hill – narrower bands indicate steeper terrain. These plots indicate the potential highest outcome (the top of the hill) and potential routes uphill, which may be different depending on one's starting point. For example, in Fig 4c, the highest point on the contour plot is in the upper right-hand corner, where patient influence activation probability and shared knowledge half-life are greatest. The terrain is relatively flat on the left-hand side of the plot: Increasing values of shared knowledge half-life at low values of patient influence activation probability (e.g., less that 0.45) results in minimal elevation change. The terrain is considerably steeper on the right-hand side of the plot: Those same changes in shared knowledge half-life, at higher levels of patient influence activa- tion probability (e.g., greater than 0.80), result in dramatic change in elevation.

### Simulated experiment

Fig 5 shows median patient health status after 5 years as a function of PVP (low, medium, high) and ER (low, medium, high) for each of the three LHN cases – selection_efficiency_maximum and evaluation_accuracy_minimum_praxis both set to low (Fig 5a), medium (Fig 5b), and high (Fig 5c) levels. In Fig 5a, PVP and ER make little difference in the outcome. In Fig 5b, PVP (medium and high vs low) and ER (high vs low and medium) both increase health status, and there is a large increase in health status between high ER (vs medium and low) in the low PVP condition. In Fig 5c, there are ceiling effects for most combinations of PVP and ER. After 5 years of simulation in a closed cohort, most patient agents, under most conditions, are experiencing high levels of health.

## Discussion

We have expressed the heuristic mid-range theory of LHNs as a specifiable, reproducible, and explicit model, instantiated in an ABM of a generalized LHN. We identified a parsimonious set of theoretically important parameters related to model outcomes of interest, showed how those parameters could be used to develop hypotheses about LHN behavior, and illus- trated how the model could inform approaches for improving LHN outcomes.

One cited shortcoming of a heuristic theory is under-specification of important constructs [33]. ABMs (and models in general) require the modeler to specify and operationalize constructs that might be otherwise ill-defined. Here, for

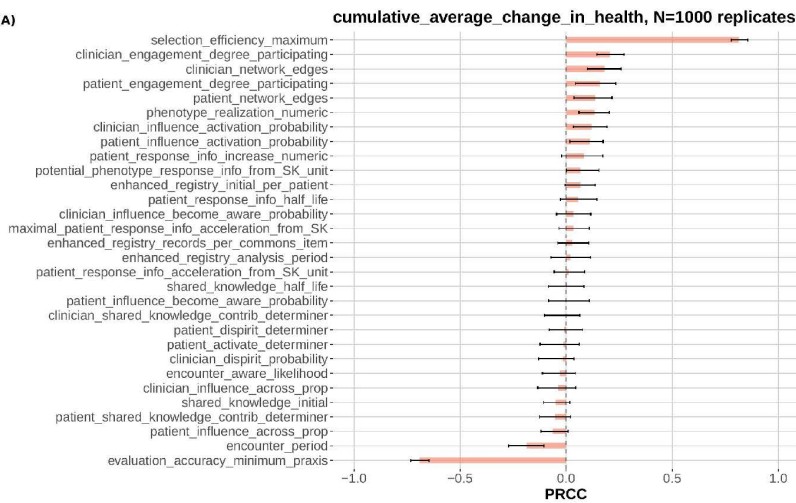

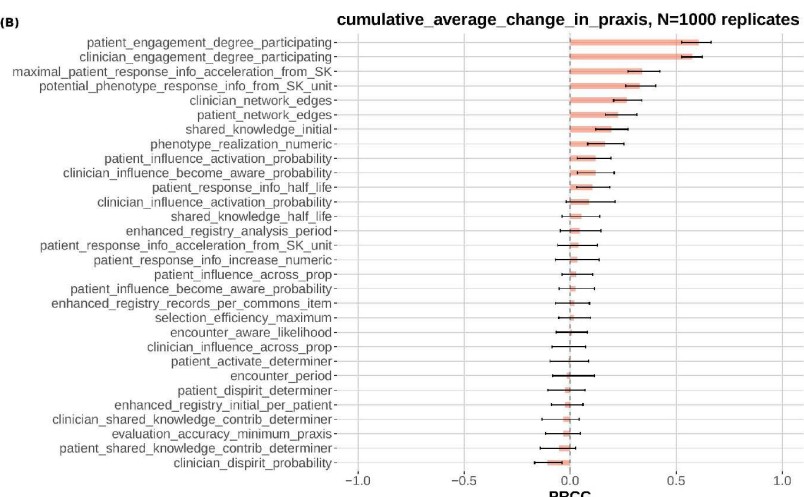

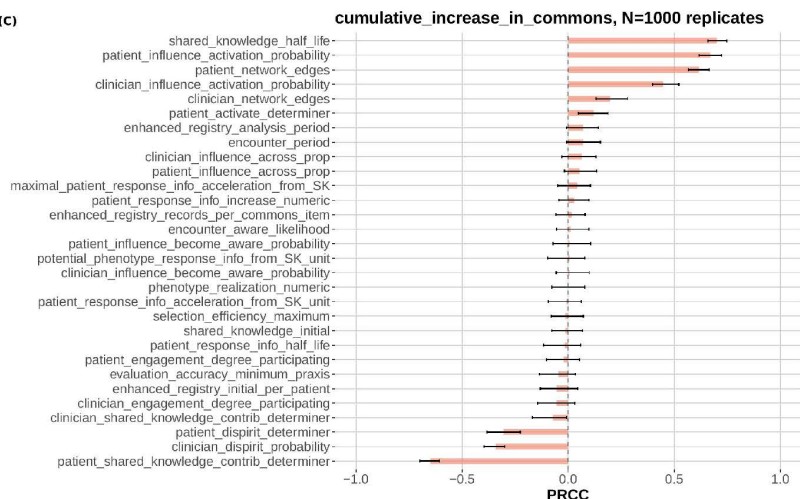

**Fig 2. Partial rank correlation coefficients (PRCCs) for health (a), praxis (b), and knowledge (c).** PRCCs take on values between −1 and +1; PRCCs with absolute values close to unity indicate strong impact on the model output. Using Latin hypercube sampling, 1000 random samples from uniform or triangle distributions of the 30 model parameters were generated. Model outputs corresponding to this set of parameters were obtained and PRCCs were calculated using the pcc function in the sensitivity library in R version 4.5.1. Bonferroni-corrected 95% confidence intervals computed using 100 bootstrap replicates.

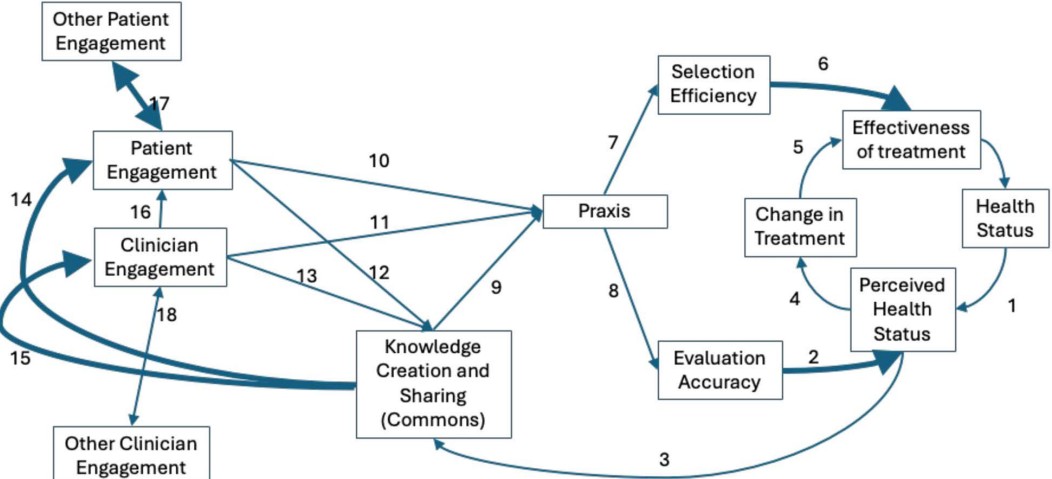

**Fig 3. High-level influence diagram for LHN ABM with arrows corresponding to largest PRCCs thickened (Arrows 2, 6, 14, 15, 17).** In the sensitivity analysis predicting health status, the two parameters with the highest PRCCs are the ability of patient and clinician agents to accurately judge whether a treatment had improved the patient agent's health status (Arrow 2) and their ability to effectively choose an efficacious intervention (Arrow 6). These parameters are both products of praxis. The two parameters with the highest PRCCs predicting praxis have to do with the amount of phenotype response information and individual response information accessed by clinicians and patients at various levels of engagement (Arrows 14 and 15). These depend on levels of engagement and on the amount of knowledge in the commons. The amount of knowledge in the commons has to do with how slowly that knowledge goes out of date (not shown in Fig 3) and the probability that a more highly activated patient will cause a less activated patient to become more activated (Arrow 17).

example, the heuristic theory of learning health networks suggests that engagement, defined as involvement in the work of the LHN, is important [9,51] as a mechanism of action. In the LHN ABM, engagement is further specified as an agent state and the effects of those states are mathematically operationalized, for example, as contributing to the commons, using information that might be in the commons, or affecting the engagement level of other agents. Translating a theoretical construct like 'engagement' into a mathematical model can lend insight into the construct's role as a mechanism of action [9]. The LHN ABM illustrates how engagement, so defined, could have an impact on the amount of knowledge in the commons, and the degree to which agents use that knowledge to create praxis, which in turn influences health status.

Translating heuristic theory into a mathematical model such as the LHN ABM may also be useful in suggesting which constructs may be important for further examination. Our model, based on the heuristic theory of LHNs, included 30 parameters thought to be theoretically important. In the present work, we investigated the sensitivity of three different outcomes to these 30 model parameters and found that those outcomes were broadly insensitive to many model parameters. For health status and praxis, there were two parameters that were clearly impactful. For knowledge, there were three. Thus, the ABM identifies a small number of parameters that might make large differences in outcomes. Fig 3 suggests how these parameters influence the system as a chain of effects leading to health status. In this model, the two most influential parameters affecting health status are the ability of patient and clinician agents to accurately judge whether a treatment had improved the patient agent's health status and their ability to effectively choose an efficacious intervention. In the ABM, these parameters are both products of praxis. The two most influential parameters affecting praxis have to do with the amount of phenotype response information and individual response information accessed by clinicians and patients at various levels of engagement. These depend on levels of engagement and on the amount of knowledge in the commons. The amount of knowledge in the commons has to do with how slowly that knowledge goes out of date (not shown in Fig 3) and the probability that a more highly activated patient will cause a less activated patient to become more activated.

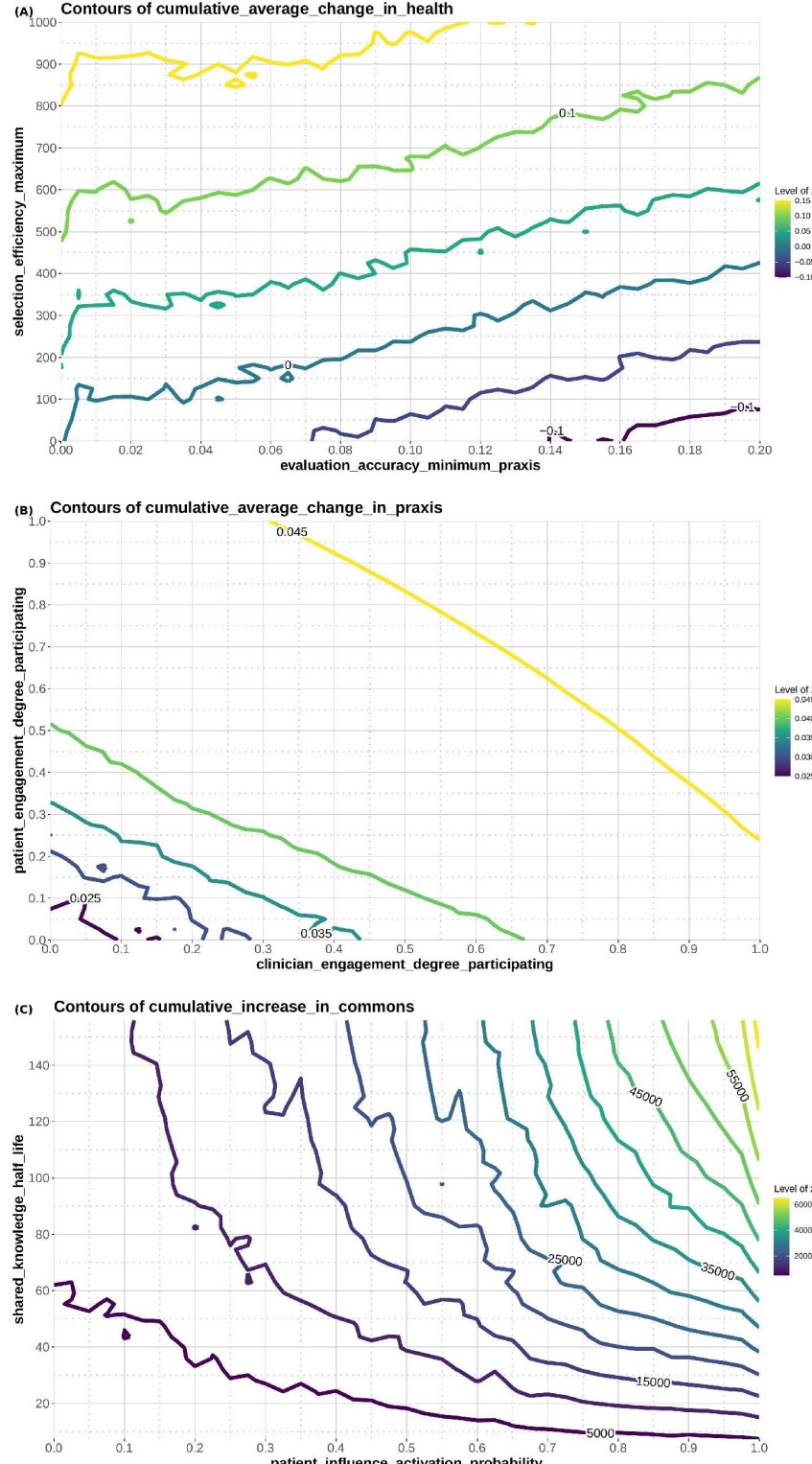

**Fig 4. Contour plot for health status as a function of selection_efficiency_maximum and patient_engagement_degree_participating (a), for praxis as a function of clinician_engagement_degree_participating and patient_engagement_degree_participating (b), and knowledge as a function of shared_knowledge_half-life and patient_influence_activation_ probability (c).** For each outcome, each of the two parameters are

divided into 41 equal increments to create an XY grid and plotted the outcome of these 1,681 runs on the Z axis (height). All other parameters were held constant at their midpoints. Contour lines joined points of equal Z value to represent Z values of constant height. The resulting contour plots represent the three-dimensional space that allows visualization, for any combination of the two parameters of not only the Z value at that point, but also the 'terrain' between the current Z value and the highest Z value.

Tracing such a chain of effects facilitates development of hypotheses about a variety of interventions that might improve health status. What if, for example, the LHN were able to improve the accuracy of evaluating treatment effects, increase engagement, or improve the amount of knowledge extracted from the commons? We show this in the contour plots, which further explores the effects of these high impact parameters. Tracing curves of uniform value illustrates the impact of the two highest impact parameters on each outcome of interest. Which intervention should be picked in a given situation will depend, for example, on factors such as the cost and ease of varying one or both of the independent variable parameters. Nonetheless, we suggest that utilizing a model such as the one described here enables the evaluation of potential interventions to further investigate and potentially pilot.

Another cited shortcoming of heuristic theory is a tendency to overlook the impact of context [33]. The factorial experiment illustrates how important context – in this case the current functioning of the LHN – might be. Fig 5a represents an LHN in which the system is producing worse health status. In that situation, PVP and ER make little difference in the outcome. Fig 5b is in a region of the contour plot where the LHN is producing stable health status. In that situation, PVP (medium and high vs low) and ER (high vs low and medium) both increase health status, and there is a large increase in health status between high ER (vs medium and low) in the low PVP condition. Fig 5c is in a region of the contour plot where the LHN is producing improved health status. In that situation, there are ceiling effects for most combinations of PVP and ER. After 5 years of simulation in a closed cohort, most patient agents, under most conditions, are experiencing high levels of health.

These results suggest that modeling may have practical implications for guiding the optimization of LHNs. Presently, heuristic theory only allows LHN leaders to determine strategy based on general rules-of-thumb, anecdote, and implicit assumptions about how LHNs work. While theory and evidence exist regarding a scientific approach to LHNs, careful exploration of mechanisms of action, in context, has heretofore been absent. The LHN ABM translates theory into explicit definitions, rules, and relationships, creating a framework for learning.

Not all interventions are equally easy to implement, nor equally efficacious. How would LHN leaders decide which to execute? Contour plots of the sort created here could provide guidance. Like navigating on terrain, it is important to know your location (current state), in what direction your objective lies, and the steepness of the terrain. For example, in this model, an LHN that has some clinician engagement and little patient engagement (e.g., (x,y) coordinate (0.25, 0.01) on Fig 4b) might increase praxis by making a relatively small increase in patient engagement. On the other hand, that same increase in patient engagement may not have as large an effect on praxis if clinician engagement is already very high (e.g., (0.50, 0.01) or (0.75, 0.01)).

Similarly, the factorial experiment suggests that the ABM can provide practical guidance. Pre-visit planning is a standard component of the Wagner Chronic Care Model, which is widely implemented [52–54]. Pre-visit planning is effective in a broad range of chronic conditions, including but not limited to inflammatory bowel disease [55], juvenile rheumatoid arthritis [56], and diabetes [57]. Although registries are common and crucial infrastructure for LHNs (the 'medium' level in the ABM), a fully realized enhanced registry, where data are captured frictionlessly in the clinical encounter, aggregated seamlessly, and are available instantly to guide clinical decision making (the 'high' level in the ABM), is far less common, despite extensive investment in informatics.

How should LHN leaders weigh their options when deciding what time and money to invest in pre-visit planning and registry technology? The ABM suggests that current LHN functioning – in this case, how effective the LHN currently is

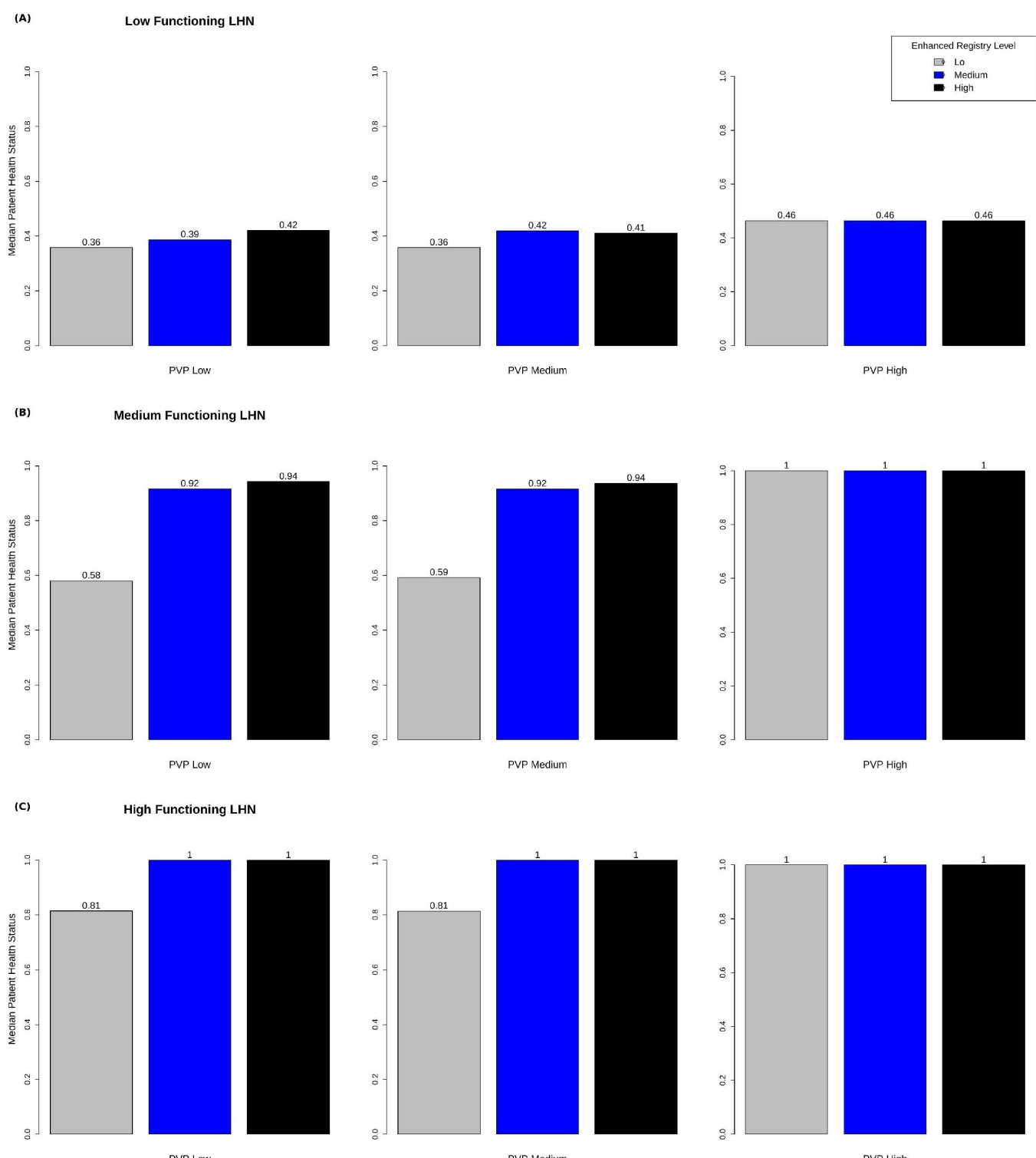

**Fig 5. Patient health status, as a function of PVP (low, medium, high) and ER (low, medium, high) for each of three LHN cases, in which selection_efficiency_maximum and evaluation_accuracy_minimum_praxis both set to low (a), medium (b), and high (c) levels.**

in improving health within a certain time period – make a difference. If the LHN is in the unfortunate situation of producing worse health status, the ABM would suggest working on ways to 'climb the hill' towards better health status before investing in pre-visit planning or an enhanced registry. If the LHN is producing stable health status, it might be efficient to improve pre-visit planning, with even greater effects for a fully realized enhanced registry – though this might prove prohibitively expensive. If the LHN is reliably improving health, modest improvements in pre-visit planning create meaningful improvement and might even render moot the need for a fully realized enhanced registry; this is an important point, given the cost and time required to create a sophisticated registry.

To the best of our knowledge, this study is the first ever ABM simulation of LHNs to investigate the mechanisms by which LHNs might produce their effects. Our study simultaneously considers the impact of choices made in defining elements of the heuristic theory (e.g., encoding 'engagement' in the way we describe and associated parameters) and LHN-level interventions. For LHN leaders, this research provides a framework for systematically thinking through the many strategies for improving their LHN and points to a more nuanced approach, based on theory-based modeling and simulated experiments, to action. While the actor-oriented architecture scheme describes the general theory for building and optimizing LHNs, it is not always clear which interventions, in which circumstances, could bring about or optimize this scheme. With a heuristic theory, the best answer to "What (set of) intervention(s) will optimize this LHN?" is "It depends." While correct, "It depends" is inadequate for effective action and risks a scattershot approach, wasted effort, and suboptimal outcomes. Absent a theory-based and empirically supported understanding of their mechanisms of action and an appreciation of the degree to which different interventions in different contexts might lead to improvements, the sheer number of options and variables would make it virtually impossible to effectively improve LHNs. By translating LHN theory into an ABM and illustrating how such a model might be useful in exploring mechanisms of action and informing potential strategy for growing and optimizing LHNs we take the first step in progressing from "It depends" to "It depends on X and Y, under Z circumstances." For LHN researchers, this research, and agent-based modeling more generally, provides a starting point for better exploration of mechanisms of action, a tool for developing and testing hypotheses, and a way to further represent and enhance a theory for LHNs.

This study can be contextualized in the broader literature. Our study suggests that, like other complex adaptive systems, LHNs can be subject to study to understand their mechanisms of action and the contextualized impact of interventions to improve their functioning. Further, this study suggests a tool that can be used in LHNs for ongoing sensing and responding [58] in leading complex adaptive systems. Much has been written on the impact of PVP and the potential of big data. This study adds to that literature by parameterizing these interventions and suggesting possible effect size, singly and in combination, under various starting conditions. We have previously described LHNs and their theoretical basis. There are several descriptive studies of LHNs over time and the multiple interventions employed to achieve improved outcomes. This study adds to that literature by suggesting the impact of theory-based mechanisms of actions for LHNs and the impact of LHN-level interventions designed to impact those mechanisms.

### Limitations

This study has several limitations. The model is necessarily an oversimplification of the phenomenon under study. It is generalized and does not capture the nuances of any one particular disease. The model is based on math that may not represent how model entities and parameters interact in the real world. The existing mid-range heuristic theory of LHNs is not quantitative and therefore is unable to render reproducible, quantitative predictions of how LHNs should behave in different situations, react to different shocks or stimuli, etc. Moreover, LHNs today are not generally sufficiently instrumented to produce data that would enable strong empirical validation for a quantitative theory. One of the functions of models is to highlight gaps–data that, if collected, are likely to result in progress in our understanding of the system being modeled and investigated. Our ABM is a first step in the field of the quantitative study of LHNs toward a reproducible, quantitative theory, albeit one based on the existing heuristic understanding of these systems.

Empirical validation is required. This would require validating inputs to ensure that the initial conditions, parameter estimates, and underlying distributions are representative of LHNs. It would require validating processes such as agent decision making and interactions to ensure that, for example, the model's representation of decision making during the clinical encounter reflects actual decision making. It would require validating outputs – both assessing how well the model outputs represent features of existing LHNs and assessing how well model predictions correspond to real-world outputs. Model outputs could also be extended to include others of relevance, such as cost-effectiveness, access, and equity.

### Future studies

Future studies will be required to use data from agents and parameters from actual LHNs to address a variety of disease processes. A potentially fertile avenue of investigation is to further build out sub-models having to do with aspects of the ABM. For example, how might social network structure, strength of ties, or the type of resources exchanged along the edges affect engagement? How might we begin to describe and model other patient agent states? As this body of research progresses, it will almost certainly be necessary to revisit and revise this model, and to develop further guidance for LHN leaders on how to use this tool and interpret results, and to account for other factors (e.g., health conditions for which limited treatment choices are available). As LHNs are instrumented and data on their functioning become more available, studies designed to help health system leaders use the ABM to develop and implement actionable interventions will be useful.

### Conclusion

LHNs can be studied using tools available from complex systems science. Developing and testing tools that guide refining and scaling LHNs is necessary to more rapidly and reliably improve clinical outcomes. This ABM adds to the theory and science of LHNs by translating heuristic theory into a computational simulation and demonstrating how it could help guide LHN leaders and practitioners to scale and optimize LHNs.

### Supporting information

**S1 Supplement. Overview, Design concepts, Details (ODD) protocol for LHN ABM.**
(PDF)

**S2 Supplement. LHN ABM parameters varying in the sensitivity analysis.**
(PDF)

**S3 Supplement. Model Syntax.**
(PDF)

**S4 Supplement. R Code for sensitivity analysis.**
(PDF)

### Acknowledgments

We are grateful for input from our colleagues Peter A. Margolis, MD, PhD., Alexandra H Vinson, PhD, Adam C Carle, PhD, Susan C. Cronin, PhD.

### Author contributions

**Conceptualization:** Michael Seid, David Bridgeland, Christine L Schuler, David M Hartley.

**Data curation:** David Bridgeland, David M Hartley.

**Formal analysis:** Michael Seid, David M Hartley.

**Funding acquisition:** Michael Seid, David M Hartley.

**Investigation:** Michael Seid, Christine L Schuler, David M Hartley.

**Methodology:** Michael Seid, David Bridgeland, David M Hartley.

**Project administration:** Michael Seid.

**Software:** David Bridgeland.

**Validation:** David Bridgeland.

**Writing – original draft:** Michael Seid, David M Hartley.

**Writing – review & editing:** Michael Seid, David Bridgeland, Christine L Schuler, David M Hartley.

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
