## [Editor Report · Decision Letter 0]

5 Mar 2025

Dear Dr. Seid,

Thank you for submitting your manuscript to PLOS ONE. After careful consideration, we feel that it has merit but does not fully meet PLOS ONE’s publication criteria as it currently stands. Therefore, we invite you to submit a revised version of the manuscript that addresses the points raised during the review process.

There are limitations in this agent-based model of Learning Health Networks, including inadequate empirical validation, poor presentation with unreadable figures, and a disconnect between theoretical optimization and practical implementation. The paper's narrative is disjointed with abrupt transitions, while its analytical approach fails to adequately explain why certain parameters are influential or how healthcare providers could translate the model insights into actionable strategies; these ultimately limit its utility despite ambitious aims.

To expedite the review process and ensure the paper meets the highest standards, I suggest revising it according to the comments below before resubmission.  This proactive step will likely save time on a round of reviews, and might facilitate a faster final acceptance.

We look forward to receiving your revised manuscript.

Kind regards,

Shrisha Rao, Ph.D.

Academic Editor

PLOS ONE

Journal Requirements:

3. Thank you for stating the following financial disclosure: [MS and DMH were funded by a grant from CCHMC entitled: A computational model health system to improve the design and implementation of Learning Health Networks and through a grant from the State of Ohio, Ohio Development Services Agency, Ohio Third Frontier, Grant Control No. TECG2022-1691.].

Please state what role the funders took in the study. If the funders had no role, please state: "The funders had no role in study design, data collection and analysis, decision to publish, or preparation of the manuscript.

4. Thank you for stating the following in the Competing Interests section: [I have read the journal's policy and the authors of this manuscript have the following competing interests:

MS and DMH are co-inventors on "A computation model of learning networks." Assignee: Cincinnati Children's Hospital. US Patent application filed 5/5/21 number: 17/291,401].

5. Thank you for stating the following in the Acknowledgments Section of your manuscript: [Funding from this project came from the State of Ohio, Ohio Development Services Agency, Ohio Third Frontier, Grant Control No. TECG2022-1691.   The content of this publication reflects the views of the authors and does not purport to reflect the views of the Ohio Development Services Agency.

We are grateful for input from our colleagues Peter A. Margolis, MD, PhD., Alexandra H Vinson, PhD, Adam C Carle, PhD, Susan C. Cronin, PhD.]

Please remove any funding-related text from the manuscript and let us know how you would like to update your Funding Statement. Currently, your Funding Statement reads as follows:  [MS and DMH were funded by a grant from CCHMC entitled: A computational model health system to improve the design and implementation of Learning Health Networks and through a grant from the State of Ohio, Ohio Development Services Agency, Ohio Third Frontier, Grant Control No. TECG2022-1691.].

6. We note that your Data Availability Statement is currently as follows: [All relevant data are within the manuscript and its Supporting Information files.]

7. Please amend either the abstract on the online submission form (via Edit Submission) or the abstract in the manuscript so that they are identical.

Additional Editor Comments:

This paper introduces an agent-based model (ABM) for studying and optimizing Learning Health Networks (LHNs), which are collaborative systems designed to improve healthcare outcomes. The authors translate social and organizational theories of collaboration into a computational model to explore mechanisms by which LHNs produce their effects. Through sensitivity analysis, the authors identify key parameters that influence health outcomes, knowledge sharing, and decision-making capacity. The model examines how pre-visit planning and enhanced registry implementation affect health outcomes under different starting conditions. The authors suggest this model provides a framework for systematically analyzing LHN optimization strategies, moving from general "it depends" answers to more specific guidance based on initial conditions and contextual factors.

The paper is not mature for publication, for the following reasons.

1. Methodological Limitations

(a) The model relies heavily on theoretical constructs without sufficient empirical validation. On page 22, the authors acknowledge that "empirical validation is required," but do not outline how such validation would be accomplished.

(b) The ABM parameters are numerous (30 parameters mentioned in the sensitivity analysis) with little justification for their selection beyond theoretical alignment. This risks creating a model that fits preconceived notions rather than reflecting reality.

(c) The authors claim that their model does not require empirical calibration (p. 5, lines 100-102), which contradicts standard modeling practice where validation against real-world data is essential for establishing model utility.

2. Presentation and Clarity Issues

(a) The flow of the paper is uneven and the narrative feels disjointed, with abrupt transitions between sections and insufficient connection between the theoretical background and model implementation.

(b) Many diagrams and figures are unreadable in their presented form. Figure 2a, 2b, and 2c showing PRCCs are particularly problematic as the parameter names are illegible, limiting their interpretability and usefulness.

(c) Figure 1, the influence diagram, contains numerous interconnected elements without clear explanation of their relationships, making it difficult for readers to understand the model's structure.

(d) Technical concepts like "praxis" are used extensively but inadequately defined (p. 9, lines 176-177), hindering comprehension of the model's core mechanisms.

3. Limited Practical Applicability

(a) While the authors claim the model helps move from "It depends" to more specific guidance (p. 7, line 125), the actual guidance provided remains abstract. For example, the contour plots (Figure 3) require significant interpretation without clear implementation steps.

(b) The model focuses on theoretical optimization rather than practical implementation. There is limited discussion of how healthcare providers would translate the model insights into concrete actions.

(c) The authors acknowledge numerous limitations (p. 22-23, lines 484-501) that fundamentally question the model's utility, including its generalized nature, mathematical representation, and lack of disease-specific factors.

4. Analytical Shortcomings

(a) The sensitivity analysis identified influential parameters, but the paper lacks a robust explanation of why these parameters are influential or how they interact within the system.

(b) The factorial experiment (Figure 4) shows different outcomes based on starting conditions, but there is limited discussion of the mechanisms producing these differences or how to determine which starting condition applies to a particular healthcare setting.

(c) The model appears deterministic in its conclusions despite the highly complex, stochastic nature of healthcare systems, raising questions about its applicability to real-world scenarios.

---

## [Author Response · Author response to Decision Letter 1]

1 Jun 2025

Response to Reviewers

PONE-D-25-08428 R1

Dear Dr. Rao;

Thank you for your review of “An agent-based model to advance the science and practice of collaborative learning health systems.” As a result of your review, we have recast and refined our objectives to better specify the scope of the manuscript. We were not clear enough that existing theory for learning health networks is heuristic and that the main contribution here is to translate that heuristic theory into a specifiable, reproducible formal logic. We have made several changes to the manuscript to indicate this refined scope:

• We have changed the title of the manuscript from “An agent-based model to advance the science and practice of collaborative learning health systems” to “An agent-based model to advance the science and practice of collaborative learning health systems.”

• The second paragraph of the introduction now states: “In this study, we take the first steps in addressing this gap by translating mid-range LHN theory into a computational model so as to ready the field for further studies and model validation. We summarize the mid-range theory of LHNs as complex systems that use an actor-oriented organizational architecture to facilitate networked coproduction. We translate that theory into an agent-based model and describe the model and underlying logic. We examine the model by computationally validating a set of theoretically relevant parameters and by illustrating an approach to simulating the effect of different interventions on possible future states for a LHN, given a set of starting conditions.”

• In the introduction, we elaborate on the individual and organizational-level heuristic theory for LHNs.

• In the introduction, we outline the limitations of heuristic theory and the relative advantages of a formal logic.

• Having a computational model of learning health networks is a large step forward for the field and a requisite for empirical calibration. We now state that “empirical calibration is an important validation-supporting process that will be addressed in future work” (lines xyz).

• We have changed the section ‘What this study adds.’ It now reads: While the above mid-range, heuristic theories of coproduction of care and the AOA describe agent motivations and behaviors and the organizational form of a LHN, critical gaps in our knowledge remain. First, while theory suggests that good care is the result of productive interactions between prepared, proactive clinicians and active, engaged patients, how does coproduction of care arise from agent actions and interactions? Second, descriptive accounts suggest that LHNs use a set of social [12,13] and technological [14] infrastructures and implement a set of processes [15] over several phases of development to achieve an actor-orientation [8,16]. How do LHNs become, or become more actor-oriented? This study begins to close these gaps in our knowledge by translating these heuristic theories into a specifiable, reproducible, and explicit model.

Below, we present the point-by-point response to review:

• Completed

• Corrected

3. Thank you for stating the following financial disclosure: [MS and DMH were funded by a grant from CCHMC entitled: A computational model health system to improve the design and implementation of Learning Health Networks and through a grant from the State of Ohio, Ohio Development Services Agency, Ohio Third Frontier, Grant Control No. TECG2022-1691.].

Please state what role the funders took in the study. If the funders had no role, please state: "The funders had no role in study design, data collection and analysis, decision to publish, or preparation of the manuscript.

• We have amended the funding information and have included the Role of Funder statement in the manuscript and the cover letter.

4. Thank you for stating the following in the Competing Interests section: [I have read the journal's policy and the authors of this manuscript have the following competing interests:

MS and DMH are co-inventors on "A computation model of learning networks." Assignee: Cincinnati Children's Hospital. US Patent application filed 5/5/21 number: 17/291,401].

• We have updated the Competing Interests statement in our cover letter.

5. Thank you for stating the following in the Acknowledgments Section of your manuscript: [Funding from this project came from the State of Ohio, Ohio Development Services Agency, Ohio Third Frontier, Grant Control No. TECG2022-1691. The content of this publication reflects the views of the authors and does not purport to reflect the views of the Ohio Development Services Agency.

We are grateful for input from our colleagues Peter A. Margolis, MD, PhD., Alexandra H Vinson, PhD, Adam C Carle, PhD, Susan C. Cronin, PhD.]

Please remove any funding-related text from the manuscript and let us know how you would like to update your Funding Statement. Currently, your Funding Statement reads as follows: [MS and DMH were funded by a grant from CCHMC entitled: A computational model health system to improve the design and implementation of Learning Health Networks and through a grant from the State of Ohio, Ohio Development Services Agency, Ohio Third Frontier, Grant Control No. TECG2022-1691.].

• We have removed the funding-related text from the manuscript and include updated and corrected information in the cover letter.

6. We note that your Data Availability Statement is currently as follows: [All relevant data are within the manuscript and its Supporting Information files.]

• We confirm that our submission contains the raw data required to replicate the results of the study. We have amended the data availability statement as follows: “All relevant data are within the manuscript and its Supporting Information files. This includes the implementation of the model in Python and the complete list of parameter values and initial conditions that are necessary for reproducing the results depicted in the study. The model is fully stochastic and therefore slight statistical variation may be observed. “

7. Please amend either the abstract on the online submission form (via Edit Submission) or the abstract in the manuscript so that they are identical.

• The abstracts are now identical

Additional Editor Comments:

This paper introduces an agent-based model (ABM) for studying and optimizing Learning Health Networks (LHNs), which are collaborative systems designed to improve healthcare outcomes. The authors translate social and organizational theories of collaboration into a computational model to explore mechanisms by which LHNs produce their effects. Through sensitivity analysis, the authors identify key parameters that influence health outcomes, knowledge sharing, and decision-making capacity. The model examines how pre-visit planning and enhanced registry implementation affect health outcomes under different starting conditions. The authors suggest this model provides a framework for systematically analyzing LHN optimization strategies, moving from general "it depends" answers to more specific guidance based on initial conditions and contextual factors.

The paper is not mature for publication, for the following reasons.

1. Methodological Limitations

(a) The model relies heavily on theoretical constructs without sufficient empirical validation. On page 22, the authors acknowledge that "empirical validation is required," but do not outline how such validation would be accomplished.

- We now include the following text in the Discussion (beginning line 693): “The existing mid-range heuristic theory of LHNs is not quantitative and therefore is unable to render reproducible, quantitative predictions of how LHNs should behave in different situations, react to different shocks or stimuli, etc. Moreover, LHNs today are not generally sufficiently instrumented to produce data that would enable strong empirical validation for a quantitative theory. One of the functions of models is to highlight gaps -- data that, if collected, are likely to result in progress in our understanding of the system being modeled and investigated. Our ABM is a first step in the field of the quantitative study of LHNs toward a reproducible, quantitative theory, albeit one based on the existing heuristic understanding of these systems. Empirical validation is required. This would require validating inputs to ensure that the initial conditions, parameter estimates, and underlying distributions are representative of LHNs. It would require validating processes such as agent decision making and interactions to ensure that, for example, the model’s representation of decision making during the clinical encounter reflects actual decision making. It would require validating outputs – both assessing how well the model outputs represent features of existing LHNs and assessing how well model predictions correspond to real-world outputs.”

(b) The ABM parameters are numerous (30 parameters mentioned in the sensitivity analysis) with little justification for their selection beyond theoretical alignment. This risks creating a model that fits preconceived notions rather than reflecting reality.

- As is now explained more clearly in the paper, the complexity of the model, including the number of parameters, follows directly from the existing, mid-range heuristic theory of LHNs in the literature. The selection of parameters for inclusion in the sensitivity analysis is based directly on theoretical alignment. Rather than prune parts of the theory when translating that theory into a model, we used the sensitivity analysis to identify a parsimonious set of parameters. In the present work, we investigated the sensitivity of three different outcomes to model parameters and found that those outcomes were broadly insensitive to many model parameters. We did not express the corresponding parsimonious models, electing instead to elaborate the implications of an ABM tool based on the existing heuristic theory.

- To address this point, we added the following to the discussion (beginning line 565): "Translating heuristic theory into a mathematical model such as the LHN ABM may also be useful in suggesting which constructs may be important for further examination. Our model, based on the heuristic theory of LHNs, included 30 parameters thought to be theoretically important. In the present work, we investigated the sensitivity of three different outcomes to these 30 model parameters and found that those outcomes were broadly insensitive to many model parameters. For health status and praxis, there were two parameters that were clearly impactful. For knowledge, there were three. "

(c) The authors claim that their model does not require empirical calibration (p. 5, lines 100-102), which contradicts standard modeling practice where validation against real-world data is essential for establishing model utility.

- We appreciate that validation against real-world data is essential for establishing model utility. We have specified our approach to validation at this stage of model development (beginning line 385): “Following Collins et al, here we define validation as the process of determining if a model adequately represents the system under study for the model’s intended purpose [43]. The model’s intended purpose at this stage is to establish and assess hypotheses, based on existing heuristic theories, about the general behavior of LHNs in consideration of key theorized mechanisms of action, and under a range of initial starting conditions and configurations. Therefore, we approached validation by a) determining the degree to which a parsimonious set of theoretically important parameters are related to model outcomes of interest and b) how those parameters could be used to develop hypotheses about LHN behavior. Further, we c) illustrate how the model could inform an approach for improving LHN outcomes.”

2. Presentation and Clarity Issues

(a) The flow of the paper is uneven and the narrative feels disjointed, with abrupt transitions between sections and insufficient connection between the theoretical background and model implementation.

- We have edited extensively to create smoother transitions and to better connect the theoretical background and model implementation.

(b) Many diagrams and figures are unreadable in their presented form. Figure 2a, 2b, and 2c showing PRCCs are particularly problematic as the parameter names are illegible, limiting their interpretability and usefulness.

- We apologize and have made the figures readable.

(c) Figure 1, the influence diagram, contains numerous

---

## [Decision Letter · Decision Letter 1]

5 Aug 2025

Dear Dr. Seid,

Thank you for submitting your manuscript to PLOS ONE. After careful consideration, we feel that it has merit but does not fully meet PLOS ONE’s publication criteria as it currently stands. Therefore, we invite you to submit a revised version of the manuscript that addresses the points raised during the review process.

The reviewers agree that the work reported in the paper has value, but have made suggestions for relatively minor improvements to the presentation that should help clarify things for readers.

We look forward to receiving your revised manuscript.

Kind regards,

Shrisha Rao, Ph.D.

Academic Editor

PLOS ONE

Journal Requirements:

Additional Editor Comments:

The reviewers are positive about the paper but have made relatively small suggestions for improvement.

Reviewers' comments:

Reviewer's Responses to Questions

**Comments to the Author**

Reviewer #1: (No Response)

Reviewer #2: (No Response)

2. Is the manuscript technically sound, and do the data support the conclusions?

Reviewer #1: Partly

Reviewer #2: Partly

3. Has the statistical analysis been performed appropriately and rigorously?

Reviewer #1: Yes

Reviewer #2: Yes

4. Have the authors made all data underlying the findings in their manuscript fully available?

Reviewer #1: Yes

Reviewer #2: Yes

5. Is the manuscript presented in an intelligible fashion and written in standard English?

Reviewer #1: Yes

Reviewer #2: Yes

Reviewer #1: (No Response)

Reviewer #2: The manuscripts presents better following incorporation of reviewer feedback and will be a useful resource for health service planners. However, there are yet some areas which could be improved/refined. More details/rationale on the selection of the 30 parameters would be helpful. Also, clear implementation steps for healthcare services/systems to implement the ABM would be useful. The authors could also explore additional outcomes such as cost effectiveness, equity..tec so as to broaden the model's utility.

**Do you want your identity to be public for this peer review?** For information about this choice, including consent withdrawal, please see our Privacy Policy

Reviewer #1: No

Reviewer #2: No

---

## [Author Response · Author response to Decision Letter 2]

22 Aug 2025

Response to Reviewers

PONE-D-25-08428_R2

Dear Dr. Rao;

Thank you and thanks to the reviewers for these suggestions to improve “An agent-based model to advance the science of collaborative learning health systems.” Below, we detail our responses to the specific critiques.

Reviewer 1:

Line 384: Does this ABM operate under a single iteration, or produce summary results from multiple iterations? The mention of replicates in the sensitivity analysis implies the latter, but I believe the manuscript would be strengthened with an explicit clarification.

• The model incorporates stochasticity at each timestep, making it necessary to compute averages and ensembles of model runs. The replicates in the sensitivity analysis relate partially to this stochasticity and partially to ensuring adequate sampling of the distributions of the varying parameters. We have clarified this beginning on line 301: “Incorporating stochasticity at each time-step of the model reflects the role of chance in actual systems and makes it necessary to analyze ensembles of model runs.”

Additionally, if multiple iterations were used, a convergence study to demonstrate that the number of iterations used is sufficiently stable would be highly beneficial.

• Thank you for pointing out this important omission. We adjusted the number of replicates in the sensitivity analyses based on stability and statistical significance of the resulting PRCCs, finding that the confidence intervals and point estimates of the PRCCs became static after 1000 replicates. We have clarified this beginning on line 473: “Trials demonstrated PRCC stability for 800-900 replicates; we used 1000 replicates for the results described below.”.

Line 532: While this section does mention the figure with relevant data, the text does not describe it in any way. Even if the simulated experiment’s results are largely repetitive, they should be nonetheless described in the manuscript so that the discussion of Figure 5 has something to refer to.

• We have added text, beginning on line 609, to describe Fig 5a, b, and c.

Lines 325/498/503/526/538: While all figures are present and readable, most of them lack a descriptive figure legend to make them independently understandable.

• We have added descriptive figure legends to Fig 2, Fig 3, Fig 4, and Fig5.

Line 436: Please add an R/Rmd file to the submission that contains the code used for this analysis and other components of the manuscript that used R.

• We have included the snippet of R code used to generate the PRCCs and corresponding CIs in the Supplementary Materials (ABM of CLHS Supplement 4 R code for senstitivity analysis.pdf).

Line 453: Please add a table of the factorial experiment’s parameters to simplify explaining how low/medium/high levels specifically differ from each other.

• This is now Table 2, starting line 515.

Parameter Low Medium High

LHN Functioning (settings for selection efficiency and evaluation accuracy that result in LHN effect on health status of…) sel eff: 10

eval acc: 0.001 sel eff: 100

eval acc: 0.01 sel eff: 1000

eval acc: 0.01

Pre-visit Planning (the maximum possible effective individual response information and the phenotype response information) 0.2 0.5 1.0

Enhanced Registry (an ordinal scale denoting on/off for automated data upload and real-time PVP) No automated data upload Automated upload but no real-time PVP Automated upload and real-time PVP

Figure 2A-C: Is it possible to provide p-values of the sensitivity analysis? The authors should also consider if it is appropriate to apply a Bonferroni Correction to this analysis.

• Using the pcc function in the R sensitivity library, we computed the PRCC 95% confidence intervals based on 100 bootstrap replicates, applying Bonferroni correction. We have updated Figure 2 and added the following text to the Figure 2 caption: “Bonferroni-corrected 95% confidence intervals computed using 100 bootstrap replicates.”.

Figure 4: Please increase the grid tick/grid line intervals to make it easier to find specific coordinates mentioned in the manuscript.

• We have added additional tick marks and gridlines, and cleaned up related axis properties, to improve readability.

Figure 5: Figure 5 shows some blurriness, but it is unclear if that is simply due to the review PDF compressing it or if Figure 5 needs to be uploaded in a higher resolution.

• We have re-created this figure to improve readability.

Line 163: Please state the language and version used to create the model in the manuscript, as well as packages used.

• We have added the following to the Methods section beginning on line 415: “The model was written in Python version 3.10 utilizing the following libraries: pandas (version 1.4.1), scipy (1.9.0), transitions (0.8.11), mesa (0.9.0), flask (2.1.3), flask-restful (0.3.9), and filelock (3.8.0). The PRCC analysis was performed using R, version 4.5.1 utilizing the sensitivity library.”

Line 209/210: double-bracketing is slightly jarring to read; can the sentence be restructured at all to only need a single set?

• This sentence now reads “…better coproduction [11] (in the ABM, praxis, explained below) between…”

Line 587/589: This sentence should end in a question mark.

• This sentence now ends in a question mark.

Line 690/711: I advise giving the limitations and future studies sections their own sub-header to better structure the discussions section.

• We added these sub-headings

Figure 5: The practicality of using Excel to create bar-charts quickly is entirely understandable, but if you have time available I would recommend recreating them in same tool as Figure 3/4 for visual consistency.

• We have re-created this figure to improve readability.

Reviewer 2:

The manuscript presents better following incorporation of reviewer feedback and will be a useful resource for health service planners. However, there are yet some areas which could be improved/refined.

More details/rationale on the selection of the 30 parameters would be helpful.

• We have added text regarding the rationale for developing the set of parameters described in Table 1. (Line 381) “We attempted to design parameters that could parsimoniously describe variation in LHNs relevant to the heuristic theory of the chronic care model and the actor-oriented organizational architecture. These 30 parameters are only a subset of all the possible influences on an LHN or of all the ways that an LHN could vary. We did not include variables characterizing broader macro-systems, such as cost-effectiveness population demographics, or payor mix.”

• We describe the selection criteria for the sensitivity analysis: (Line 428) “Parameters included in the sensitivity analysis were developed from the theoretical elements of LHNs, as described above and in Table 1. Other parameters in the model were developed to enable simulation of different health conditions (e.g., number of phenotypes, number of treatments, severity, course, relapse probability) or to simulate different initial conditions (e.g., patients joining the network over time vs already all enrolled, patients entering and leaving the cohort vs closed cohort). These parameters were held constant in the sensitivity analysis.”

Also, clear implementation steps for healthcare services/systems to implement the ABM would be useful.

• Thank you for this suggestion. We have added text to the Future Studies section regarding using the ABM to develop and implement actionable interventions (Line 811).

The authors could also explore additional outcomes such as cost effectiveness, equity, etc., so as to broaden the model's utility.

• We agree that these additional outcomes would broaden the model’s utility and have included this suggestion in limitations (Line 798).

---

## [Editor Report · Decision Letter 2]

26 Aug 2025

An agent-based model to advance the science of collaborative learning health systems

PONE-D-25-08428R2

Dear Dr. Seid,

We’re pleased to inform you that your manuscript has been judged scientifically suitable for publication and will be formally accepted for publication once it meets all outstanding technical requirements.

Kind regards,

Shrisha Rao, Ph.D.

Academic Editor

PLOS ONE
---

## [Editor Report · Acceptance letter]

PONE-D-25-08428R2

PLOS ONE

Dear Dr. Seid,

I'm pleased to inform you that your manuscript has been deemed suitable for publication in PLOS ONE. Congratulations! Your manuscript is now being handed over to our production team.

Kind regards,

on behalf of

Dr. Shrisha Rao

Academic Editor

PLOS ONE